# "Kauaka e kōrero mō te Awa, kōrero ki te Awa: An Awa-Led Research Methodology" *(Don't Talk about the Awa, Talk with the Awa)*

**Tom Johnson** †

Whakauae Research Services Ltd., 19 Ridgway Street, Whanganui, Aotearoa 4500, New Zealand;
tom@whakauae.co.nz

† Mōkai Pātea Nui Tonu, Ngāti Hauiti, Te Āti Haunui-a-Pāpārangi.

**Abstract:** Indigenous people continue to develop methods to strengthen and empower genealogical knowledge as a means of conveying histories, illuminating current and past values, and providing important cultural frameworks for understanding their nuanced identities and worlds across time and space. Genealogies are more than simply a record of a family tree; they are a rich tapestry of ancestral links, representing a tradition of thought and connection to entities beyond the human. This article proposes an Iwi-specific methodological approach to conducting research based on the specific paradigms (ontological and epistemological) of Māori (Indigenous peoples of New Zealand) from the region of Te Awa Tupua in the North Island of Aotearoa, New Zealand. A Whanganui world view can be actioned as an operating system within research by developing a bespoke place-based methodology drawing on kōrero tuku iho (ancestral wisdom) to conduct research amongst a genealogical group with whakapapa (genealogical connection) to a distinct geographic locale. This methodological shift allows the inclusion of human research participants and more-than-human, including Te Awa Tupua (an interconnected environment around the Whanganui River) and Te Kāhui Maunga (ancestral mountains that feed the Whanganui river) as living ancestors. Whanganui ways of knowing, doing, and being underpin a worldview that situates Te Awa Tupua and tāngata (people) as inter-related beings that cannot maintain their health and wellbeing without the support of one another.

**Keywords:** environmental ancestor; kōrero tuku iho; indigenous ancestral connection; whakapapa methodology; Whanganui; Te Awa Tupua

## 1. Introduction

Whenua (land) and whakapapa (ancestry) represent an inseparable nexus forming Māori concepts and experience of community, home, and identity (Boulton et al. 2021). Whakapapa is a genealogical tool with multiple reference points for relationality with the rest of existence (King et al. 2022). As interdependent and complementary components woven together in the natural world, whakapapa is the sacred thread that connects humans deeply amongst all other species, including time, space, and the spiritual and cosmic realms (Mead 2016; Te Rito 2007). In whakapapa, Māori weave a robust and deep whāriki (woven mat) of ancestral connections, introducing themselves with geographical ties to environmental forebears like mountains or rivers (Te Aho 2019). While there are variations in worldviews between Iwi, and the diversity of each individual tribal member further contributes to these variances, a Māori worldview typically shares an intricate connection to taiao (the local natural environment), integrating layers and histories involving more-than-human entities (Marsden and Royal 2003; Mikaere 2011; Smith 1999; Te Rito 2007; Williams 2004). As physical embodiments of taiao, mountains, rivers, and natural phenomena form the foundation of an Iwi's genealogical connection to the land.

Whakapapa Māori to taiao is reinforced when Iwi use kōrero tuku iho (ancestral wisdom) to share place-based wisdom about their environments, shaping unique Iwi and hapū identities. Kōrero tuku iho are physical manifestations of history, identity, culture, and intellectual traditions that tether to a people's ever-evolving sense of self and act as psychic anchors (Te Awekotuku and Nikora 2003) to an enduring intimate familial relationship with taiao. Kōrero tuku iho come from Tūpuna accounts achieved through deep contemplation and interaction with our environments (Walker 1990); thus, our pūrākau (stories) include Māori philosophy, epistemology, cultural norms, and beliefs (Lee 2009). Our tales provided the context for understanding our Iwi narratives (Walker 1992) and were employed as a paradigm for communicating and teaching our philosophies and values—our worldview. While Iwi narratives may exist in written form, the oral renditions employed by Tūpuna for knowledge transmission, constituting a living history, were and continue to be conveyed face-to-face across generations (Mahuika 2012, 2019). These oral traditions are not mere subjects found in books; kōrero tuku iho transcend mere textual subjects. They embody narratives directly witnessed and heard from living and deceased individuals, as well as the woven universe of more-than-human ancestors, making their presence tangible and familiar. Here is where the metaphors of whakataukī come into play: "they help us talk about intricate and complex relationships—things we simply cannot convey through linear, verbal expressions" (Cajete 2015, p. 207). Whakataukī/whakatauākī are a type of kōrero tuku iho, succinct messages which encapsulate and encode Tūpuna intelligence and transfer this to the living. Understanding the metaphorical world and giving it space to emerge in various contexts is essential for organizing and using Indigenous knowledge (Cajete 2015). Particularly important is the development of examples where kōrero tuku iho can be used in contemporary settings, a space between two paradigms (Indigenous and Western) that situates "the life worlds of contemporary Indigenous people in the dynamic space between ancestral and western realities" (Yunkaporta and McGinty 2009, p. 58). Whakataukī/whakatauākī are used in a conceptual methodology in Section 6.

Each Iwi (tribe) and hapū (subtribe) have unique expressions of their genealogical taiao connection: carefully coded oral taonga (treasures) disseminated in the form of waiata (songs) (Haami 2022), mōteatea, oriori (including ancestral chants, laments), haka (ceremonial dance) (Ka'ai-Mahuta 2010) pūrākau (stories) (Lee 2009), whakataukī/whakatauākī (proverbs), and other tangible Tūpuna (ancestral) tools able to carry the unique contexts of the Indigenous experience of belonging to living ancestral landscapes. The title of this article is a well-known kōrero tuku iho to Whanganui Iwi, which codes a philosophical directive to descendants of Whanganui Iwi: not to simply speak *about* their ancestral river but to speak *with* it. I draw from this whakatauākī to center how I might conduct place-based research in a way that makes sense within a Whanganui point of view.

> "As our pāhake have explained, we are defined by our ancestral mountain, our ancestral rivers, and our ancestral land. They are the source of our wellbeing—spiritually, intellectually, and physically. We do not separate our wellbeing from [their] wellbeing. . . Nor can we possess them. They do not belong to us—we belong to them".

(New Zealand Environment Court and New Zealand Legal Information Institute 2004, p. 29)

This article promotes the incorporation of Indigenous epistemic and ontological perspectives in researching with our communities, emphasizing the inclusion of more-than-human ancestors. The intent is to offer a template for Indigenous researchers, prompting reflection on the cultural narratives they inherit (kōrero tuku iho) and how their epistemological paradigms can shape research projects that prioritize serving their communities rather than fitting into Western academic frameworks. The article starts with how Iwi (tribes) in my papakāinga (homeland) and neighbouring Iwi regard nature as an ancestor, as evidenced in their kōrero tuku iho. This is followed by examples of international Indigenous communities' experience of environmental genealogical connection and their ability to commune with taiao. Thirdly, a short exploration into how voicing taiao in a

tribally oriented, place-based methodology in health research could be useful in conducting research amongst Indigenous communities, and the considerations researchers might take in developing their own. Concludingly, a conceptual methodology employed in a PhD project investigating the wellbeing practices and rituals performed by Māori men of Te Awa Tupua is presented to illustrate its practical application.

## 2. Whakapapa to Taiao

Over many generations, our Tūpuna (ancestors) migrated from Hawaiiki, the ancestral home of Māori (the Indigenous people of Aotearoa (New Zealand)), across East Polynesia. They brought with them a deep understanding of taiao, a repository of expert environmental knowledge that enabled them to successfully navigate the Pacific. As they made Aotearoa home, this knowledge was tweaked and adapted, finely tuned to the local conditions, and a strong sense of connectedness to place was amplified by an oral memory culture (kōrero tuku iho) weaving Māori into the landscapes they inhabited through stories (Smith 2020). Although Māori worldviews vary from Iwi to Iwi, a common thread is the importance placed on ancestry in both the material and ethereal realms, where sites of environmental significance, recognized as Tūpuna (ancestors), are consistent with previous Indigenous epistemological writings that acknowledge the significance of its more-than-human relatives, referring to ancestors in the mountains, mists, rivers, rocks, plants, and the unseen (Mika 2017; Smith 2020). These natural features have familial and genealogical links to humankind (Warbrick et al. 2023a).

For Ngāti Rangi, an Iwi of the central North Island, Mount Ruapehu (Matua Te Mana) is referred to as 'grandfather', a treasured ancestor (Potaka-Osborne et al. 2018) who is "an active, living entity" (Gabrielsen et al. 2017, p. 471). Comparably, The Waikato River embodies the mana and mauri (vital life energy) of the Waikato-Tainui people, who regard it as a revered Tūpuna (ancestor) (Muru-Lanning 2007, 2016; Salmond et al. 2019). Ngāi Tūhoe (an Iwi of the North Island) attribute their whakapapa descent to both their sacred mountain Maungapōhatu and the Atua (deity) Hinepūkohurangi (the mist that envelops the Te Urewera region) (Te Awekotuku and Nikora 2003). With a thorough understanding of the local environmental spaces they made home, Tūpuna left to their descendants this legacy of place-based oral mātauranga (traditional knowledge, wisdom, and cultural insights of Māori) as a vital component of a larger, more comprehensive awareness of the natural and spiritual world (King et al. 2007). Through Kōrero tuku iho, Tūpuna were able to express these more-than-human genealogical origins in a repository of coded cultural knowledge, an intergenerational bank of mātauranga-Iwi (Iwi-specific knowledge), which includes a distinct worldview informed by the natural environments that they exist in.

Whanganui Iwi regularly use the term "Te Kāhui Maunga" to express whakapapa in Maunga (mountains) in the centre of Te Ika-a-Maui (the North Island plateau). These Maunga include Ruapehu (Matua te Mana), Tongariro (Matua te Toa), Ngauruhoe (Matua te Pononga), Taranaki (Matua te Tapu), and Pihanga (Matua te Hine). Each Maunga has an individually significant whakapapa to local Iwi and hapū (e.g., Some hapū refer to Ruapehu as "Te Potae-o-Te-Atihaunui-a-Paparangi" (Simon 1986), yet collectively, Te Kāhui Maunga are drawn upon by Whanganui Iwi when introducing themselves by the whakataukī "E rere kau mai te Āwanui, Mai i te Kāhui Maunga ki Tangaroa" (The great river flows from the mountains to the sea. I am the river; the river is me). Pepeha (a traditional form of introducing oneself) conveys one's cultural identity and ancestry and establishes upfront the speaker's many relations, human and more-than-human, with earth, mountains, and rivers. The rivers originating in Te Kāhui Maunga are metaphorically compared to an umbilical cord which anchors the tribe to the very soul of its forebears (Wai 1130 2013) back to Paerangi-o-Te-Whare-Toka (the keeper of the fires of the mountain tribe of Ruapehu) (Wilson et al. 2022). So significant is Whanganui Iwi connection to taiao, ancestors manifest physically, mentally, biologically, and genetically, demonstrating a clear epistemological human–nature relationality and their connection to sentient ancestors. This whakapapa exists along an infinite interconnected time and spatial plane, where Māori can live in

a limitless present, able to recall Tūpuna in their numerous forms (including taiao) and interact with them at any given moment (Smith 2020). Viewing taiao as an entity imbued with the accumulated knowledge, experience, and wisdom of previous generations, Iwi of Whanganui engage with taiao as a sentient being with its own mind, feelings, and free will, and they treat it accordingly with the same respect they might show a grandparent, teacher, or elder.

For this article, I use the term "Te Awa Tupua" to reference the river and the entities (land, fish life, water tables, stories, Tūpuna, source mountains) as one inseparable entity (Tupua Te Kawa n.d.), and I do not list the hapū and whānau who whakapapa here, which is for *them* to know. Whilst the Whanganui Awa is metaphorically named "Te Taura-whiri-a-Hinengākau" (the plaited rope of Hinengākau), which hints at the multi-layered understandings of ancestral worlds (Wilson et al. 2022), another Tūpuna name is Te Wainui-a-Rua or Te Awanui-a-Rua (the great river of Rua Tipua) a central ancestor to the Whanganui-Taranaki region. Te Awa Tupua is a manifestation of a distinct mauri (life force) and holds the role of a living ancestor with spiritual integrity for the Whanganui Iwi. The metaphorical significance of water as their lifeblood establishes a medium for meaningful communication (Wai 167 1999). Supported by the Whanganui whakatauākī, "He ripo, he tipua, he kāinga" (*"At each rapid, kaitiaki and people dwell"*), it elucidates the ancestral genealogical relationship within waterways and mountains, emphasizing a direct communion rather than mere discourse about the Awa. Taiao has a voice (Wai 167, 1999), and a Maunga can be characterized as *"A forgotten elder. A quiet Tuakana. An uncited source"* (Koroi 2021, p. 18). In Whanganui, two humans are appointed by Iwi and the Crown to act and speak on behalf of the more-than-human ancestor Te Awa Tupua (Mika and Scheyvens 2022). The function of He Pou Tupua (the name of this role) serves as a paradigmatic instance of Whanganui epistemological logic, a practical application of Kōrero tuku iho which accentuates critical junctures at which nonhuman vitalities intersect, amalgamate, and collaboratively shape human world-making. This analytical lens accentuates the salience of interactions between humans and more-than-human entities in shaping perceptual constructs of reality within the Whanganui worldview.

Whakapapa extends to trees, rocks, birds, and waters, recognized as expert Tuakana (older family members) who have enduring knowledge of the land. This knowledge is voiced, as recounted by Cheryl Smith (2020), who reveals a historical continuity of Tūpuna teachings in Whanganui where the winds, water, clouds, birds, mist, and rain "constantly spoke to us" (Smith 2020, p. 24). Cheryl Smith recalls these ancestral conversations, given that she was raised at the end of a generation that still held Tūpuna teachings in practice (Smith 2020). Pā McGowan (2020) and Gabrielsen et al. (2017) echo this sentiment, under-scoring that land "talks back" to humans in a way that is constantly evolving in response to the diversity of the landscape. "Koro Ruapehu is constantly changing. Sometimes he's sleeping, sometimes he's active—sometimes he erupts" (Gabrielsen et al. 2017, p. 463). Wooltorton et al. (2022) remind Indigenous peoples that recognizing and greeting our relations in the forms of living nature, rivers and multispecies beings is transformative, and that our river relations "have never forgotten us though. They keep calling, waiting patiently- as they always have" (p. 394). M. Jackson (2020) assures us that it is never too late to restore that ancestral relationship, for "in whakapapa no relationship is ever beyond repair" (M. Jackson 2020, p. 140), and a whakapapa connection is an inviting relationship, a constant engagement with taiao (Heke 2016). These dynamic and reflexive interpretations underscore the ongoing revitalization of kōrero tuku iho, which form the foundation of Māori epistemologies and worldviews.

Like our ancient ancestors, we can all feel, see, and hear with the wisdom of our rivers (Wooltorton et al. 2022). Rivers and other geological sites in nature are also sacred living ancestral beings amongst Aboriginal Australians (Hattersley 2009; S. Jackson 2005; Wooltorton et al. 2017; Wooltorton et al. 2022), where to "sing a place" affirms their custodianship and intimate relationship with the land, flora, and fauna (Kohen 2003). The Yellowknives Dene (or Weledeh) dialect of Dogrib is exemplified by Coulthard (2010), who uses the

word "land" (dè) to refer to a concept that encompasses not only the land but also humans, wildlife, rocks, plants, lakes, rivers, and other natural elements. From this vantage point, humans are as much a part of the earth as any other living creature and are not the only entities in this web of interconnections to be endowed with some sort of immaterial soul or volitional control. The conversational powers of rivers are expressed in other Indigenous cultures, including the Pessamiulnuat, Innu of Pessamit people of Canada, Indigenous communities in America (Fox et al. 2017; Manikuakanishtiku et al. 2022), and Sub Saharan Africa (Mamati 2018; Maseno and Mamati 2021). Places and rivers are constantly engaged in intuitive communication and messaging, which may be decoded or revealed through animal messages such as the arrival and departure of birds, the direction of the wind, intuitions, and feelings (Wooltorton et al. 2022). The human–taiao relationship is expressed in the long-lasting bonds between tāngata (people) and whenua (land), and Indigenous cultures share the view that the natural world is a living being, and Indigenous narratives emphasize human ancestry in and genetic connection to the natural environment.

### 3. Whakapapa to Taiao and the Academy

Māori whakapapa (genealogical connection) to the environment is well theorized, but in the literature, I found little evidence of how this connection might be *put into action* inside of a research methodology. Perhaps this is due to Indigenous ways of understanding and engaging with genealogical meaning, knowledge, and practice being marginalised in academic discussions on genealogy. Historically, Māori communities have experienced ingrained problems with research, where the exploitation of their intellectual property led to a distrust of researchers, their techniques, motivations, and methods. Numerous Indigenous peoples have been subjected to the theft of their knowledge, cultural artifacts, imagery, and tales, which were then fed back to them through a foreign Western lens and presented as truth. Non-Indigenous researchers frequently lack sufficient comprehension to draw accurate conclusions regarding the lived realities of Indigenous peoples (Mutch and Wong 2008). The lack of culturally relevant research approaches, argue Macfarlane and Macfarlane (2019), is a reflection of the persistent inability to comprehend Indigenous cultures. To this day, non-Western forms of knowledge, especially those that emphasize the intangible and incalculable components of the human experience, are discounted by the so-called Western impartiality (Cartier 2020; Hikuroa 2017). The Westernised academy suppresses and damages generations of Indigenous knowledge (Hikuroa 2017; Smith 1999) and continues to marginalize Indigenous perspectives as inferior, "non-scientific" or superstitious nonsense (Coburn et al. 2013; Cooper 2012; Watts 2013), or in the context of Aotearoa, position Māori people as the "other" (Boulton 2020; Pihama 1993).

Māori understand how whakapapa to taiao plays a major role in shaping health and wellbeing (Panelli and Tipa 2007). Economic stability, tribal identification, health, and spiritual grounding are all built on the notion of belonging to and connection with taiao rather than ownership and control (Durie 1998, 2001; Hutchings and Smith 2020; Te Aho 2011). Recognizing and empowering these Māori constructions of meaning not only enriches cultural heritage but also positively influences health and wellbeing outcomes (Durie 1998; Walker 1990). Durie (2006) said, "human wellbeing is inseparable from the natural environment" (p. 12) and is a fundamental tenet of Indigenousness, indicating wellbeing as multidimensional and interdependent in relationship to all forms of life (Mika 2017; Pere and Nicholson 1991). Health, in fact, *is* the environment (Rereata Makiha cited in (Warbrick et al. 2023a). Understanding these indigenous perspectives is essential for developing culturally competent healthcare approaches, potentially reducing health disparities and improving health outcomes among Māori populations (Ministry of Health 2019; Cormack et al. 2018; Reid et al. 2019). A Whanganui worldview understands the universe as an orderly, dynamic system based on living and non-living phenomena and an Awa and Maunga, which are inseparable from its people (Wai 167 1999). Where wellbeing research is concerned, the persistent exclusion of Indigenous knowledge in peer review

publications and policy development reflects a failure to uphold Indigenous sovereignty and self-determination in the research process.

Given the inherent interdependence of health and wellbeing with taiao and recognizing the imperative to cultivate more trustworthy Indigenous research practices, this article proposes the utilization of Iwi (tribally oriented) place-based methodologies in research. Place-based research prioritizes locally informed tactics, ideologies, tikanga, and future ambitions, reflecting the phenomena of the enduring ancestral and wellbeing-oriented ties Māori have with taiao. A deliberate shift toward Iwi (tribally oriented) methodologies may be a more responsive and responsible approach to research where they are able to be tailored to a particular group of individuals, their rohe (region), their hapū (sub-tribe), addressing their needs, and aligning with their worldviews. This intentional shift aims to transition from research being perceived as something "done to" Māori to an approach where research is held, guided, and led by Māori, encompassing both tāngata (people) *and* taiao (environment). If cultural belonging provides Māori with insulation against the harsh realities of life in a colonized environment (Bennett and Liu 2018; Matika et al. 2017), then a deeper connection to the specific sites and kōrero tuku iho that Iwi Māori draw this belongingness from may produce a more responsive research methodology. A bespoke, place-based approach to study honors ancestors in taiao by focusing on the influence of specific characteristics of a location and positioning the Indigenous populations that live there as the experts, recognizing their skill in having monitored, observed, interpreted, cared for, and lived in congruence with taiao and the wellbeing of both.

In the context of a Māori wellbeing-focused PhD project, I crafted a methodology to investigate the rītenga (rituals and practices) of Tāne Māori (Indigenous Māori men) with whakapapa to Te Awa Tupua (Whanganui River area). The study aimed to understand how these individuals maintain and manage their wellbeing. This qualitative research drew from Whanganui-specific kōrero tuku iho, including pūrākau (stories and narratives), tohu taiao (environmental observations), whakataukī/whakatauākī, interviews, grey literature, ruruku (incantation), tātai (short, codified chants directed to the environment), and kōrero-a-Iwi (tribal-specific teachings) to identify the sources of strength which nourish Tāne Māori. The research aims to amplify wellbeing narratives and shift away from the deficit discourse of Māori men (Johnson 2021) by elevating the diverse voices of Tāne Māori and fostering alternative, strength-based, and resilient-focused ways of making sense of Māori men's wellbeing. This necessitated a consideration of how taiao functions centrally in the way Tāne Māori uphold wellbeing and conceptualize the idea of wellbeing within their Whanganuitanga (practices indigenous to the Whanganui region) to build the evidence-base required for change. The next section considers the process of revitalizing the tāngata–taiao relationship, as guided by kōrero tuku iho, into a methodology for conducting research.

## 4. Considerations in Developing an Iwi Methodology

The methods employed in this methodology aim to inform a movement from a broad-stroke "homogenised" Māori approach to research toward Iwi-specific methodologies. Before conceptualizing the methods, I asked my Awa and my Tuakana how might we improve our mutual wellbeing from what we learn on this journey. I reflected; perhaps the *"how to"* interview of an ancestral Maunga or Awa in this article is not as important as questions of why. If, within a Māori epistemology, *the purpose of indigenous knowledge is not merely to describe the world (acquire facts about phenomena) but ultimately to understand how one may live well in it"* (Royal 2009, p. 114), then in my Indigenous research, I must carefully consider devising respectful methods to collect, interpret, share, and contextualize knowledge by integrating research approaches within my specific cultural context (Hermes 1998). If the scholarly literature fails to adequately capture the worldview of Tāne Māori in my local context (Johnson 2021), exploring indigenous theorizing becomes imperative for research methodologies that are locally grounded and context-specific. This encompasses recognizing and understanding their lived experiences with nature and entities beyond the

human realm. To affirm Tāne Māori autonomy, providing a platform for Taiao (nature) and Tūpuna (ancestors) to articulate their teachings becomes pivotal within this framework.

Royal (1998) claims whakapapa as a research methodology or instrument suitable for analyzing natural occurrences, origins, links, interconnections, and even predicting the future. A whakapapa methodology is a tool that explores "the nature of phenomena, the origin of phenomena, the connections and relationships to other phenomena, describing trends in phenomena, locating phenomena and extrapolating and predicting future phenomena" (Royal 1998, p. 4). Epistemologically, a methodology is an opportunity to reconstruct Indigenous epistemologies as an operating system within science and research that allows researchers to connect with their own culture, explore their past and present, and construct meaning on their own terms (Mila-Schaaf and Hudson 2009). What can be known; what knowledge is genuine, legitimate, and useful; what are the underlying assumptions for what can be known; and who are the knowledge holders, both human and more-than-human, are all questions that epistemology seeks to answer (Andersen and Walter 2013; Smith 1999). Indigenous knowledge is seen as "belonging to the cosmos", and we humans are only the "interpreters" of that knowledge (S. Wilson 2008, p. 38). In this section, I attempt to interpret some publicly available kōrero tuku iho from my rohe, explaining the process of decoding ancestral teachings into guiding principles embedded within a research project (discussed in Section 6). Additionally, I offer considerations for researchers developing their own place-based methods.

Similar processes are taking place in other research methodologies, with many tribes desiring to know what "their tribal" research methodology is, much as tribal groups desire to relate to their own ecosystems in terms of health (Heke 2016). Amongst hapū groups, there are distinct perspectives and experiences that have been influenced by context and experience, providing a diverse spectrum of lived Māori realities (Durie 1998; Durie 2001, 2003, 2012), meaning that the level of depth of information and comprehension that each group member holds will differ (Durie 2001; Houkamau and Sibley 2015). Each Iwi has its own core principles linking them to their environmental context, and this unique tribal application of principles and values cannot be applied by another tribe as they will have their own (Doherty 2012). Each Iwi and hapū has its unique definitions and local applications of values, and it would be naïve to suggest that one method would homogenously apply to all Māori (Hapeta and Palmer 2014). Applying a simplified, generic, and hegemonic "Māori" methodology to this research project would not honour the unique epistemological view and Mātauranga-a-Iwi (specific Iwi knowledge) of Whanganui men. An example of such a methodology is "Piupiu as an identity framework", a Tūhoe-centric model rooted in Tūhoetanga which "determines the origins, nature, methods, and limits of Tūhoe knowledge according to Tūhoe" (Fraser and NZQA 2012).

When the natural landscape is imbued with ancestral qualities, re-thinking and re-orienting how a living and indivisible whole, which comprises physical and metaphysical elements, is required to consider how a river and Mountain as a Tuakana (older sibling) and Tūpuna may lead and guide us and can be active participants in research. In an article reflecting on research methods in the use of generating mātauranga Māori or Indigenous knowledge, (Tuhiwai Smith et al. 2016) ponder, *"Perhaps we, as indigenous scholars, struggle to find the right terms to use to articulate something we know and care for, respect and remember, and that we seek to engage with, knowing that our ancestors might be looking on, and that the next generations will ask us, 'What did you do in your time to ensure that our peoples flourished?'"* (p. 151).

I strived to leverage my unique Te Awa Tupua perspective to actively integrate conversations (kōrero) from both tāngata (people) *and* taiao (environment) into the research project. Te Awekotuku and Nikora (2003) explain that "people and places derive their identities from each other to a significant extent. It is the betweenness that is important—the relationship that is created and sustained" (p. 11). Environmental history carries with it a conviction that the history of humanity and the history of the environment only make sense if explored together (McNeill 2001). The multiple relationships that indigenous

researchers have with both human and more-than-human entities in their natural settings should inform researchers' worldviews and interpersonal dynamics within the research process (Chilisa 2012). Because the entities, the seen and unseen in taiao, are not passive but active participants in the web of life, hearing these voices requires a relational framework that invites other beings into the discussion (Lowe and Fraser 2018). To recognize local, place-based epistemologies, the research process requires consistent and ongoing reconceptualization and reconsideration, which is illustrated with an example in Section 6. Intentionally applying a methodological approach within the rohe where Te Awa Tupua lives and breathes requires a research strategy embedded and implemented within and via Te Awa Tupua, one which can coordinate research activity relating to humans and more-than-humans integrating and enabling Tūpuna and taiao voice.

Privileging a Whanganuitanga lens at the ngako (core) of the research provides a guide to operationalizing all elements that comprise Te Awa Tupua in its wholeness: *"[where] the collective Earth, Sky and Waters of Te Awa Tupua and all its natural life, including its people, [may continue] coexisting interdependently as expressions of sacred life energy"* (Rāwiri 2022, pp. 1–2). Taking a place-based (Whanganui-centric) worldview follows other Indigenous post-humanist work, where the research demands a decentering of humans (Cutter-Mackenzie-Knowles et al. 2020), and so this methodology strives to reconsider what it means to be human in multispecies environments through its explicit Whanganui epistemology. Using whakapapa as the entranceway to conduct Kaupapa Māori research sets the parameters and the nature of an inquiry and allows implications of the research to be considered in the planning and ethics phase. Whakapapa has considerations beyond genealogy, and includes the ownership of material including spiritual ownership (Durie 2003), which belongs to the people and not the academy. Smith (1999) explains whakapapa is more than genealogy but also "a way of thinking, a way of learning, a way of storing knowledge" (p. 234). Engaging in place-based research with Māori requires diligent and critical planning to maintain a broader comprehension of the researcher's roles and duties in data collection and analysis within the framework of whakapapa. The introduction of Kaiponu, a unique Whanganui coding system created by Tūpuna to protect the sacred transmission of ancestral stories (kōrero tuku iho), is a key part of starting this place-based illustration.

## 5. Kaiponu: A System of Protecting Whakapapa Knowledge

The exceptional cultural relevance of Indigenous information, particularly genealogical records, and the centrality of protecting and controlling this intellectual property for Māori self-determination, cultural practices, and values meant that they retain an important place in research interaction. The importance of inter-generational knowledge to Māori is captured by Hiroa and Buck (1926), who explain how historical narratives of Māori were not idle stories but instead contained the knowledge of *"things celestial, things terrestrial and ritual"* (p. 183). This knowledge was taught in formal, intentionally constituted houses of learning (whare wānanga), which *"had an unbroken succession from ancient times"* (Hiroa and Buck 1926, p. 183). Māori modified and adapted their coding system to include information that was distinctive to each hapū (sub-tribe) and was informed by rohe (regional specifics of taiao).

In Whanganui, a system of protecting knowledge was devised known, as Kaiponu, "as a way of preservation and protection" of knowledge (Haami 2017, p. v). Knowledge holders—or as (Hiroa and Buck 1926) call them, *"keepers of the ancient traditions"* (p. 187)—preserved and safeguarded the unwritten knowledge, and to keep it safe, it was disclosed only in whare wānanga (Mahuika and Mahuika 2020). Knowledge keepers in our Māori communities hold the expertise to bridge the gap between the physical and metaphysical realms, guide spirits on their quests through time and space, bind ancient genealogies with contemporary realities, and access wisdom and insight from the generations before us (Tuhiwai Smith et al. 2016). An ethic of selective disclosure to non-Iwi venues, Kaiponu or Kaipono, is a Whanganui Iwi statement about the protecting and preserving of Whanganui

tribal proprietary information (Gray-Sharp 2021) and in enacting Kaiponu, only members of the Whanganui Iwi are allowed to know the whole whakapapa *"to protect and maintain Whanganui hapū specialist expert knowledge"* (Rāwiri 2022, p. 433). This emphasis is crucial due to historical challenges such as robbery, murder, misappropriation, and commercialization of Māori kōrero tuku iho, necessitating a strategy to protect the intellectual property of Whanganui Iwi members.

Through the deliberate use of kaiponu, research can purposefully safeguard the legacy of kōrero tuku iho of Whanganui while expanding its current applications as a research method. In the context of the Ph.D. research, participants are invited to wānanga to collectively contribute their thoughts on Kaiponu in deciding "what to do" with the information gathered in the research (what dissemination or outputs would be of most value) and with *whom* this information is shared. Not only can this provide optimal information dissemination and targeted direction of research findings, but Kaiponu underscores a participatory decision-making model, empowering the community to actively shape the trajectory and impact of research outcomes. Before initiating my research, I adopted a "pre-ethics" approach (Lowe et al. 2020), a strategy of "deeper deep listening" (p. 1) not only to taiao but also tāngata in Te Awa Tupua to understand how to enact this local intellectual tradition. As a result of many conversations and cups of tea (Potaka-Osborne 2019), this proposed methodology attempts to follow the values shared by my Tuakana (elders) and aims to conduct research which is inclusive and pono (truthful) to the worldview of the participants and can generate new knowledge from kōrero tuku iho, which serves its participants, whilst also being able to enact Kaiponu simultaneously.

## 6. Conceptual Methodology: Tupua Te Kawa—An Awa-Led Research Framework

Considering Te Awa Tupua as the primary ancestor of Whanganui Iwi, the methodology devised for this whakapapa-based research is 'led by the Awa' through conversations with it, incorporating instances where the ethical system of Kaiponu is used and where re-interpretation for contextual consideration is explained with a rationale. Four whakataukī are operationalized as a methodology, a flexible research strategy that can be viewed more as a philosophical guide than a rigid methodological tool, underpinned by a variety of Whanganui worldviews, knowledge, and traditions.

The Te Awa Tupua Act of 2017 (The Whanganui River settlement) uses Tupua Te Kawa as an organizing framework and does not favour any set of values above others; rather, it emphasizes the opportunity to rethink the connections between people, places, and authority in a way that benefits everyone. Tupua Te Kawa, the central values of Whanganui, according to Gerard Albert, *"It puts the river at the centre of the picture and asks us to organise around it"* (Speech at Ngapuwaiwaha Marae in Taumarunui 2019) (Albert 2019). I draw from 'Tupua Te Kawa', a set of whakataukī Indigenous to Whanganui which depict the genealogical order of creation and contain valuable insights into a Whanganui worldview (Salmond 2014; Te Aho 2014; Tinirau 2017; Wai 167, 1999). Tupua Te Kawa hold the innate values (Whanganui tikanga and kawa) of the region (Tinirau 2017) and articulate the whakapapa connections between human and more-than-human (Te Aho 2014). *"Methodologically, Tupua te Kawa is predicated on more-than-human and human interactions, which informs the underlying conceptualizations of the Whanganui River"* (Haami 2022, p. 37).

Kōrero tuku iho, such as whakataukī, can be used to interpret and build realities as they allow people to engage in *"sophisticated Indigenous ways of knowing"*, whereas methodologies could be *"engaging with and negotiating cultural metaphors that can express, structure and inspire thinking and learning processes"* (Yunkaporta and Shillingsworth 2020, p. 7).

Drawing from Haami (2022), whakataukī from Te Awa Tupua serve as both ethical and methodological pillars, rooted in Iwi whakapapa and central to Whanganui oral tradition. Echoing McRae's (2017) perspective on whakataukī as a holistic framework spanning across genealogies, I agree with Haami (2022) that they encapsulate past, present, and future cultural norms. Tupua Te Kawa is intentionally employed as an adaptable framework for guiding research within this tribal context, aiming to offer insights for other Iwi to

leverage their kōrero tuku iho in refining their research approaches. This approach seeks to contribute to the decolonization of Māori perspectives, challenge a homogenous view of Māori, and provide a point of reference for other Indigenous peoples to reassert the validity of their ways of knowing and being through their epistemologies.

*6.1. Ko te Awa te mātāpuna o te ora: The River Is the Source of Spiritual and Physical Sustenance*

**The orientation.** As an ongoing wānanga with Te Awa Tupua, at all stages throughout the research, *"ko te Awa te mātāpuna o te ora"* orients the research, the researcher, and participants inside of an ancestral river relationship. This orientation prioritizes the Whanganui way of life and lens on the world, building upon the epistemology of all research participants—both human and more-than-human entities intricately bound physically and spiritually in minute detail to Te Awa Tupua. The researcher, attuned to the awa and taiao, keenly observes signs and actively seeks ways to engage in a dialogue, allowing guidance to unfold from the wisdom inherent in the natural surroundings. The researcher's attunement to taiao exemplifies a proactive approach to maintaining the "right relationship". By keenly observing signs and actively seeking dialogue with the natural surroundings, the researcher navigates the intricate web of connections within the indigenous landscape.

Located and protected within ancestral sites of strength, this whakataukī clearly articulates the connection to the wellbeing of the study (the research questions) and the rītenga (rituals), which can be enacted, co-constructed, and performed to keep participants well. Rituals in taiao (such as ruruku, whakataumaha (meditations), tātai, and pure (incantations and clearing rituals for wellbeing)) will need to be observed and constructed throughout the project to keep all components (research, researcher, and participants) in "right relationship" with each other (Koroi 2021).

*6.2. E rere kau mai te Awa nui mai i te Kāhui Maunga ki Tangaroa: The Great River Flows from the Mountains to the Sea*

**The toolbox.** Using the metaphor of a river, "E rere kau mai te Awa nui mai i te Kāhui Maunga ki Tangaroa" encourages researchers to adapt their methods to the diverse physical features of the rohe and to be like a river in how it pivots and adapts its flows to the needs of taiao and the people around it.

This whakataukī also recognizes the vastness of physical and meta-physical space which exists. Whakapapa connections to Te Awa Tupua can stretch as far as Rangitīkei-Ruapehu-Manawatū and include how human participants may self-identify in relation to Te Awa Tupua, keeping this relationship open to interpretation by all who are involved in the research project. This whakataukī asserts the need to consider the sites of data collection to determine where it is appropriate to conduct interviews and wānanga (*e.g., in the context of the study where is it 'tika' or right to gather information- is it at the beach, mountain or riverside, while hunting? While in the garage?*)

With relationships, the whakataukī serves as a guide to research participant recruitment, and in the context of research, the whakataukī examines what the appropriate tools required are (*e.g., What are the methods for data collection? Are they fit for purpose to hold both human and more-than-human voice?*). It allows the researcher to consider what they need; in my case, I developed my own Maramataka journal (a decision-making tool codified on the basis of Māori ecological knowledge based on the systematic study of environmental indicators, rhythms, and cycles) (Warbrick et al. 2023b) as an active, reflective device to capture field notes which may contextualize some of the research findings. This fieldwork, or 'Awawork' journal, will be a reflective guide for me throughout my research, and like the methodology, is an iterative, experiential learning tool to integrate and render information stream (e.g., including sketches, environmental observations) in an aesthetic and embodied within Whanganui epistemology. This whakataukī is also a guide for the researcher to explore the researcher's axiology and positioning in the research—where relationships can be conceptualized as a network of connections from Mountains to sea.

*6.3. Ko au te Awa, ko te Awa ko au: I Am the River and the River Is Me*

**The Lifecycle.** This meso-macro lens to the research speaks to the life cycle of the research project from its inception and reminds the researcher that wherever the head-waters end up, they are required to ensure that the research benefits both the Awa and the communities of the Awa. In this sense, this whakataukī speaks to the origins and destinations of the knowledge generated, grounded in the need to listen, observe, and include more-than-human knowledge. This whakataukī metaphorically conceptualizes the research journey as a sustained collaborative effort fostering deep, long-lasting connections with participants and emphasizes the consequential ripple effects generated by the actions undertaken within the research project.

*Ko au te Awa* holds upfront the Whanganui view that Te Awa Tupua, its waters, and all entities connected with it can be voiced, heard, and connected with. This instructs the researcher to consider other communication methods in listening to the voice of Awa and in translating that voice appropriately to the needed audience *(e.g., Is the research noticing the environmental patterns occurring? Is it tuning with a local maramataka and considering observations? Who is it appropriate to share this insight with?).* This contributes to the critical phase of the research, where data analysis and write-ups of the findings are required. *(e.g., Has the discussion section been inclusive of more-than-human entities?).*

*Ko te Awa ko au* holds space for the interpretation of the voice, which is subjective to "who" the Awa is at that point and whether the participant has chosen to voice an ancestor. The river may provide advice (via human voice or other) in ethical considerations, critically affirming the value of guidance by a Tūpuna.

*6.4. Ngā manga iti, ngā manga nui e honohono kau ana, ka tupu hei Awa tupua: The Small and Large Streams That Flow into One Another Form One River*

**The Offerings.** This whakataukī in the research context frames up and leads all considerations relating to the dissemination and uptake of information from the research. *Does it serve the Awa and those who draw their wellbeing from it?* This whakataukī serves as a guide when the research is ready to begin the dissemination and design of outputs, as it holds an obligation to honour human and the more-than-human entities and may raise such questions as *will it create rubbish for the river. Do they advocate for the wellbeing of natural as well as human environments?*

*Ngā manga iti* (the small streams) represent new knowledge found in this research; the smaller streams are often overlooked or undervalued, considered "other" or irrelevant. The small streams are the evidence, stories, and realities of Te Awa Tupua—the precious practices—the mātauranga of Whanganui, which will be carefully woven into disseminations aligned with the practice of Kaiponu.

*Ngā manga nui* (the large streams). In the context of this methodology, Ngā manga nui are indicative of pre-existing knowledge *(What is already known?).* These encompass readily observable data, strategies, and existing systems that play a role in shaping the prevailing conditions for wellbeing. The methodology employs this facet to discern the drivers for change, strategically identifying levers that can be employed to enhance the impact of the research project. Examples of such strategic considerations involve assessing the most impactful outputs for participants and discerning the preferred style and format of output that participants deemed beneficial for their engagement.

Within the methodology, *"e honohono kau ana, ka tupu hei Awa tupua"* considers the curves and contours of Māori identity, treasuring all koha (gifts), all stories, or kōrero contributing to this project. To accommodate the variances that exist within and across Whanganui Māori communities, the research processes need to be adaptable to participant needs, just as the design of the research outputs must serve the needs of the research participants stated above in *The Lifecycle.* The variegated environment symbolises the different beliefs, worldviews, and interpretations of diverse individuals (recognized as experts of their own realities), and the research must honour and validate all Māori conventions as normal and valued for all contributions to the kumete (bowl) and contribute with deliberate

intent to the dynamic waters of Te Awa Tupua. Grounded in a fixed locale, informed by ancestral wisdom, and yet continuously moving and evolving, this methodology aims to actualize ancestral narratives, operationalize whakapapa knowledge, and prioritize the tāngata-taiao tradition from its inception and throughout the research process.

## 7. Conclusions

Taiao, as a Māori ancestor, holds significant genealogical importance in Whanganui, particularly the interdependencies between Te Awa Tupua and Te Kāhui Maunga with local Iwi. Leveraging ancient perspectives on the intricate interconnectedness of human and more-than-human life, this article advocates prioritizing rohe-specific, place-based epistemologies for tailored research methodologies with Indigenous communities. This proactive approach aims to mitigate the repercussions of neglecting diverse knowledge sources, emphasizing the need to include more-than-human perspectives. When considering taiao and culture as protective factors for Māori, research grounded in these foundations may foster more inclusive and meaningful research outputs and outcomes for Indigenous communities. This is an important act of Kaupapa Māori research against a backdrop where the mainstream rhetoric includes detractors who continually question the legitimacy and practicality of Indigenous perspectives and argue that a lack of transparency on operational methods and demonstrated benefits may diminish the credibility of a kōrero tuku iho led methodology in comparison to established Western scientific approaches.

Together with the participants, I used this place-based methodology to share kōrero, which helped us comprehend the results and create solutions that are founded on our common worldview and still provide room for other perspectives. I believe that the research participants serve as custodians of the ancestral wisdom inherent in Te Awa Tupua and practise their own kaiponu. The nuanced expressions of their varied and subtle identities require the use of bespoke methodological approaches which are able to hold diverse voices and narratives that are aligned with their unique worldview, which centres Taiao as a teacher, elder, and source of wellbeing. These approaches are crucial for obtaining meaningful responses that align with the cultural context of Tāne Māori, and they seek to provide evidence of well-being practices for the future sons of Te Awa Tupua. By utilising the Iwi-focused technique, I am able to incorporate their ongoing feedback during the dissemination process. This approach results in distinctively formed outcomes that offer valuable insights and connections that would be unattainable with a non-Iwi-focused strategy, serving as a countermeasure against deficit-focused research and challenging negative stereotypes and narratives surrounding Māori men. This positions Tāne Māori (the participants in the PhD research) as experts in their respective diverse contexts, embodying the profound traditions, values, and knowledge embedded in their experiences and expressions as descendants of a great river.

The Tupua Te Kawa framework is one such example of how using kōrero tuku iho informs a bespoke approach to the people–place divide in conducting whakapapa-based research. The methodological framework shared here is an intentional shift from "one-size fits all" conceptions of Indigenous research approaches toward valuing the unique and varied mātauranga of a singular locale. Additional inquiry is needed to explore the efforts of other Indigenous researchers who integrate their genealogical and ancestral knowledge into practical research methodologies, along with an examination of the potential outcomes arising from such approaches. This pursuit constructs a rich and abundant kumete (bowl), overflowing with resources to quench the intellectual thirst of our collective teina (younger siblings)—the next generation of emerging researchers of Indigenous communities who I hope enter more inclusive and welcoming spaces rooted in their unique worldviews. Ultimately, researchers may draw upon the guidance and wisdom of more-than-human entities by distilling ancestral wisdom and nuanced interpretations of personal observations within the contextual framework of their natural environments. *Kauaka e kōrero mō te Awa, kōrero ki te Awa.*

**Funding:** This work was supported by Whakauae Research Services under the Health Research Council of New Zealand (HRC) grant number HRC22/509.

**Institutional Review Board Statement:** Not applicable.

**Informed Consent Statement:** Not applicable.

**Data Availability Statement:** No new data were created or analyzed in this study. Data sharing is not applicable to this article.

**Conflicts of Interest:** The author declares that this study received funding from Whakauae Research Services under the Health Research Council of New Zealand (HRC). The Health Research Council of New Zealand was not involved in the study design, collection, analysis, interpretation of data, the writing of this article or the decision to submit it for publication.

## Glossary

Glossary of words included in this article (Te Reo Māori and New Zealand English).

Note on the glossary: It is crucial to acknowledge that not all Te Reo Māori words can be directly translated into English, as some require experiential understanding, contextual awareness, and a Te Ao Māori perspective for full comprehension. This nuance is especially important in academic articles, where the inherently expansive nature of Te Reo Māori may be constrained or "disciplined" by other linguistic registers. Different Iwi may employ distinct words or variations to articulate and explain concepts, reflecting the rich cultural and linguistic diversity within the Māori community. In presenting this glossary, I have endeavoured to provide definitions that align with my understanding, but for further insight, it is recommended to consult additional works in Te Reo Māori.

| Te Reo Māori | New Zealand English |
| --- | --- |
| Aotearoa | New Zealand |
| Atua | Deity |
| Haka | Ceremonial dance, or recited form of dance accompanied by action |
| Hapū | Kinship group, clan, tribe, subtribe |
| Hawaiiki | Ancestral home of Māori |
| Iwi | Extended kinship group, tribe, nation, people |
| Kaitiaki | Guardian, guardians |
| Kaiponu/kaipono | To keep to oneself, withhold |
| Kaupapa Māori | Māori centred approach. It refers to a Māori way of doing things, grounded in Māori cultural values, perspectives, and practices. |
| Karakia | Chant |
| Kōrero tuku iho | Ancestral wisdom comprised of history, stories of the past, traditions, oral traditions from Māori ancestors |
| Kumete | Bowl |
| Maunga | Mountain, mountains |
| Māori | The Indigenous peoples of New Zealand |
| Maramataka | Māori lunar calendar |
| Matua te Mana | Prestige of the father- referring to Mount Ruapehu |
| Mātauranga Māori | Refers to the traditional knowledge, wisdom, and cultural insights of Māori, encompassing a holistic understanding of the world, including language, spirituality, customs, and ancestral connections. |
| Mātauranga-a-Iwi | Specific Iwi knowledge, wisdom, and cultural insights |
| Mauri | Life force |
| Ngako | Core, the essence |
| Pāhake | A Whanganui term for elders |
| Pepeha | Pepeha is a traditional Māori form of introduction that conveys one's cultural identity, ancestry, and connections to specific landmarks or places in a concise and meaningful manner |
| Pūrākau | Myth, ancient legend, story |
| Pure | A release, a ritual ceremony to remove tapu |
| Rohe | Boundary, district, region, territory |
| Ruruku | An incantation, a Whanganui word similar to the Māori word 'Karakia'. To delve into the depths, draw it up, and share it with the world |

| | |
|---|---|
| Taiao | Refers to the natural world or environment, encompassing the ecological, spiritual, and cultural dimensions of the land |
| Tāngata | People |
| Tāne Māori | Māori men |
| Taonga | Treasure/treasures |
| Tātai | Short codified chants directed to the environment to help understand and make sense of what is happening in the natural world around you |
| Te Awa Tupua | A concept and region around the Whanganui River. Also, a unique legal Act that acknowledges the river as an indivisible and living entity with its own rights, values, and identity |
| Te ika-a-Maūi | The North Island of New Zealand |
| Te Kāhui Maunga | A collective of mountains in Central North Island, embodying the interconnectedness of five sacred peaks within Māori cultural and spiritual contexts. |
| Teina | Junior relative |
| Tikanga | Correct procedure, custom, habit, lore, method, manner, rule, way, code |
| Tuakana | Elder sibling or senior relative |
| Tupua Te Kawa | A framework of Whanganuitanga, encompassing ancestral traditions and cultural protocols specific to the Whanganui region |
| Tupuna | Ancestor (singular) |
| Tūpuna | Ancestors (plural) |
| Wānanga | Seminar, conference, learning space where knowledge is shared and passed on |
| Whakapapa | Genealogy, genealogical table, lineage, descent |
| Whakataukī | Proverb, significant saying |
| Whakatauākī | Proverb attributed to someone |
| Whakataumaha | Meditation |
| Whare Wānanga | Ancient houses of learning |
| Whāriki | A woven mat or carpet that holds cultural significance |

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
