# Peer review of "“Kauaka e kōrero mō te Awa, kōrero ki te Awa: An Awa-Led Research Methodology” (Don’t Talk about the Awa, Talk with the Awa)"

_genealogy, doi:10.3390/genealogy8010030_

Round 1

Reviewer 1 Report

Comments and Suggestions for Authors

This is a very solid article written and centered on māori ways of being, knowing, and doing. The references and citations are thorough and appropriate including numerous Indigenous scholars who are experts in the subject area and in Indigenous methodologies. Unfamiliar termonologies in the Māori language are well explained making the article centered in Māori thought languages, yet accessible to those unfamiliar with the Māori language.

I find no weakness or missing information in this article; it is one that will greatly benefit all researchers. Reading the article brought many thoughts to mind on how this article will advance research with, for, and by Indigenous communities, and place-based research. This article will most likely be widely cited by researchers in diverse fields of thought and scholarship.

There are now numerous articles and books on Indigenous methodologies. However, this article clearly brings a deeper understanding, of the importance of place-based methodologies and genealogies including human and non-human relations.

LINE 381. There seems to be a word missing at the beginning of the sentence.

 "are anchored on ancestral knowingness"

Author Response

The article has been greatly improved thanks to your professionalism and insightful comments. I really value your input and appreciate the time and thought you put into offering suggestions for improvement. Your expertise added immense value to the article, and I am grateful for the detailed and constructive suggestions you offered. 

Edits:

  • On line 381 I have amended the sentence which may have been cut short in the formatting of this document. Thank you for spotting this.

Once again, I appreciate your thoughtful participation. I hope that our academic paths will meet again in the future.

Ngā manaakitanga (kind greetings) from Aotearoa, New Zealand,

The author

Reviewer 2 Report

Comments and Suggestions for Authors

This is a fascinating and readable piece on bespoke place- and regionally-based research methodology that involves a deep connection to place and access to mātauranga Māori. It gives rich insights into Māori beliefs and practices based around whakapapa and genealogical connections. Some examples of particular research topics of relevance to your community, in addition to your own research on how Māori Tāne men manage their well being would be useful.  The justification for such a methodology and the claim in your abstract that the Whanganui world view can be actioned as an operating system for science and research would be strengthened if particular issues or problems can be identified -- as you hint at in referring to other Māori ecosystems and health (ll. 297-99) and in referring to the research questions as the well being of the project (l. 464). What other questions important to the indigenous population might lead them to engage tāngata and taiao in conversation? For example, issues related to health and well being such as how to cope with stress?  What  conditions would encourage which this methodology to be applied?   You refer to constructing a research project to serve the people (l 141); is there an example of such a project – even hypothetical -  e.g. environmental issues?  Related to this are questions of how can this methodology be a model for other communities? Can that be pinned down more specifically in terms of actual research questions and content?

A special combination of natural features in the ancestral landscape is the key inspiration in evolving this collaborative methodology.  While stressing this heritage as unique to the Whanganui region, just as the particularities of other regional landscapes determine the distinctive methodologies of other iwi, it would  be worth locating the methodology in a wider context:  have similar research methodologies been developed by Indigenous communities beyond Aotearoa, for example,  among those you name in referring to widespread Indigenous knowledge about the power of rivers (ll. 164-66)?

In the context of your own research can you expand more (l. 222) on the ‘deficit discourse’  of Māori men, such as what is consists of, how it develops and operates?  Do the diverse voices among Māori Tane men that you elevate (ll. 231-32) include voices from tupuna and taiao?  If they are one and the same, then are the diverse voices alternatives to the deficit discourse? An  indication of your research questions and outcomes would be relevant here.

l. 377 The  research is seen as  a way of storing knowledge --in Whanganui, through the protection  and ownership system called Kaiponu. The fact that  ‘new knowledge’ is not publically available raises questions of the research’s wider relevance:  are there ways in which the public and other research communities might share in and benefit by the research,  and under what the conditions  might this happen - eg by publication, open forum, community advice or wānanga?  Does this strategy for protection of intellectual property apply to your proposed research methodology (as well as your own research), and  are there occasions in which such protection will be relaxed?

You have mapped some traditional categories used in empirical research (eg research questions, data collection, data analysis, self reflective journal) onto your practical application of korero tuku iho through summarising whakataukī . This suggests your model is situated between qualitative and quantitative research (but is mainly the former). Is it pertinent to introduce other interrelated aspects such as size of sample, organisation of data for analysis and presentation of results? Is there potential for revising the questions and direction of the research based on the data collection (i.e. the reflective journal)? What examples of ‘one size fits all’  (l. 565) conceptions of Indigenous research exist, against which your model is compared? 

Comments on the Quality of English Language

There are some  incomplete or flow on sentences (ie that should be two sentences)  and some grammatical and syntactic errors

A few examples

Abstract- this sentence can be improved for sense

‘By developing....enables a Whanganui world view to be actioned’ – rephrase as ‘a Whanganui world view can be/is able to be  actioned’

l. 52, revise  sentence and rewrite ‘carrying’ as ’that carry’

ll. 66-68 revise for clarity- ‘demonstrating’ would make more sense as the present tense ‘demonstrate’

l. 73, add comma after practice (nb spelling )

ll. 82-83, Clarify that it is Indigenous people who consider Western ontology as ‘external’ and Western science as ’redundant’ – e.g. in l. 83  “are considered by Indigenous people as ’external’”

ll. 85-86. Draw out the contrast here: i.e.  it is the Indigenous ontology that is about the individual’s relationship  to reality; but (not ‘and’) the Western view that knowledge is individualised

l. 232. Could use hyphens to indicate sense units:  eg strengths-based, resilient-focussed, and l. 252 whakapapa-based

l. 242 Repetition  of phrase ‘before beginning’ – suggest delete the 2nd use

l. 322, similar kind of repetition

ll. 326-328 revise – add ‘meant that they‘  before ’retain’

l. 335  ‘planning on’;  comma splice – this  should be the beginning of a new sentence

l. 342, ‘adopted how they would code’ - revise for sense and clarity

ll. 381-3- a sentence fragment that does not connect with the article and should be deleted

l. 397, Add ‘is’ before ’rightfully’

ll. 427-29 an incomplete sentence

l. 451 participants?

ll. 475-77 clarify this sentence- add ‘that’ after ‘toolbox’

l. 491. Add ‘an’ before environment

References- some  entries are incomplete lacking publication details or page numbers,  and entire list needs proof reading for  appropriate style, e.g.

Royal 2009

Ruruka whakatupa -  Publication details ? is this in correct place?

Williams J. publication details

Wilson C.T. location of publisher?

Te Aho 2011 (why Wendy Harrex?? ) and article  in Law Review- no apparent title

Gray-Sharp  University of Waikato- is this a thesis?

Wirihana—thesis title usually in roman and in quote marks,  not italics

Presentation

Often the name of a cited author is  given in parenthesis when functioning as the subject of the sentence  eg  l. 333, ‘(Smith 1999) explains whakapapa’

Is this the style guide followed? It makes more sense to be written as ‘Smith (1999) explains whakapapa’

See also ll. 346, 338, 375

Article needs careful proof reading throughout, and checking for consistency in spelling etc

Author Response

Reviewer 2. Peer-review of : Kauaka e kōrero mō te awa, kōrero ki te awa (Genealogy Journal).

Kia Ora,

The article has been greatly improved thanks to your professionalism and insightful comments. I really value your input and appreciate the time and thought you put into offering suggestions for improvement. Your expertise added immense value to the article, and I am grateful for the detailed and constructive suggestions you offered. The article has undergone a restructuring process based on your feedback and that of the other three reviewers, incorporating the provided comments, to enhance its precision and focus.

I have made a list of the edits made below.

Once again, I appreciate your thoughtful participation. I hope that our academic paths will meet again in the future.

Ngā manaakitanga (kind greetings) from Aotearoa, New Zealand,

The author.

NB: Please note that since editing this article required combining three other peer reviewers, line numbers referenced in my responses may have changed slightly.

Comment

Adressed/response

Comments on the Quality of English Language

Edits made:

·       Abstract sentence II12 changed to “A Whanganui world view is able to be actioned as an operating system within science and research by developing a bespoke place-based methodology to conduct research amongst a genealogical group with whakapapa (genealogical connection) to a specific geographic locale.”

·       I52, revised sentence to ’that carry’

·       II.82-83 “Demonstrating” kept to illustrate the present-progressive tense (its ongoing right now)

·       I.73 comma added, spelling kept as practice (noun, NZ/UK spelling)

·       II. 82-83, added  “are considered by Indigenous people as ’external’” for clarity

·       ll. 85-86 the word ‘but’ replaces ‘and’ to emphasis the contract between Indigenous ontology and a western view of knowledge being individualised

·       l. 232 I have used hyphens to indicate sense units: strengths-based, resilient-focussed, and also on l. 252 whakapapa-based

·       l. 242 Repetition  of phrase ‘before beginning’ – deleted on the 2nd use

·       I.322 replaced with “I introduce”

·       ll. 326-328 I have added ‘meant that they‘  before ’retain’

·       I. 335  ‘planning on’;  comma splice – this  should be the beginning of a new sentence- I have broken the sentence and rewritten for sense “Engaging in place-based research with Māori requires diligent and critical planning to maintain a broader comprehension of the researcher's roles and duties in data collection and analysis within the framework of whakapapa.”

·       l. 342, ‘adopted how they would code’ – have revised to “Māori modified and adapted their coding system to include information that was distinctive to each hapū (sub-tribe), and was informed by rohe (regional specifics).”

·       ll. 381-3- a sentence fragment has been deleted

·       l. 397, have added ‘is’ before ’rightfully’

·       ll. 427-29 an incomplete sentence- I could not find this

·       475-77 I have added ‘that’ after ‘toolbox’

·       The references have been updated to “ACS” format

Some examples of particular research topics of relevance to your community, in addition to your own research on how Māori Tāne men manage their well being would be useful.  

I like this idea a lot but the wordcount already feels quite high and it seems outside of the scope of this article on composing an Indigenous whakapapa/genealogical based methodology for research. I would absolutely expand on this in another article once the findings/discussion from the PhD is ready to share. Also much Māori research of relevance to the community has followed Western conventions and is not unique to Whanganui- it follows a pattern of New Zealand health concerns which have been researched as isolated health concerns (eg. Diabetes, suicide, obesity) and none have taken an explicit Whanganui lens to the research and have not connected it to the health of Te Awa Tupua.

The justification for such a methodology and the claim in your abstract that the Whanganui world view can be actioned as an operating system for science and research would be strengthened if particular issues or problems can be identified -- as you hint at in referring to other Māori ecosystems and health (ll. 297-99) and in referring to the research questions as the well being of the project (l. 464).

Thank you for this great suggestion. I have reflected on this and removed the word ‘science’ and restructured that statement inside another section- section three of the article where considerations in developing a methodology is discussed.

What other questions important to the indigenous population might lead them to engage tāngata and taiao in conversation? For example, issues related to health and well being such as how to cope with stress?  What  conditions would encourage which this methodology to be applied?   You refer to constructing a research project to serve the people (l 141); is there an example of such a project – even hypothetical -  e.g. environmental issues?  Related to this are questions of how can this methodology be a model for other communities? Can that be pinned down more specifically in terms of actual research questions and content?

I can’t find any other environment or place-based methodologies in the Indigenous literature which use the natural environment and the ancestral knowledge attached to that environment as a conceptual framework to operationalise an Indigenous methodology. The examples I reference in the article in II 164-66 touch on the Indigenous Aboriginal worldview in relation to their rivers, yet the cited research does not develop a methodology specific to that worldview. I have had a second scan of the literature and cannot find any other international Indigenous examples to include here.

I have however moved your questions into the new article by restructuring it into four sections, and I hope I have answered this in section 3- considerations in making a methodology. I am happy to edit further if required.

A special combination of natural features in the ancestral landscape is the key inspiration in evolving this collaborative methodology.  While stressing this heritage as unique to the Whanganui region, just as the particularities of other regional landscapes determine the distinctive methodologies of other iwi, it would  be worth locating the methodology in a wider context:  have similar research methodologies been developed by Indigenous communities beyond Aotearoa, for example,  among those you name in referring to widespread Indigenous knowledge about the power of rivers (ll. 164-66)?

Great idea, I have added these examples as suggested into section2, ll180-220.

In the context of your own research can you expand more (l. 222) on the ‘deficit discourse’  of Māori men, such as what is consists of, how it develops and operates?  Do the diverse voices among Māori Tane men that you elevate (ll. 231-32) include voices from tupuna and taiao?  If they are one and the same, then are the diverse voices alternatives to the deficit discourse? An  indication of your research questions and outcomes would be relevant here.

This is a great question and I thank you for this. I need to get more clear in my explanations of this particular phenomenon. What I understand is that they are one and the same, as nature is a part of us, thus, whilst the deficit discourse may come from humans, there could be value in rising the alternative narratives through other vehicles including the more-than-human. Because of the restrictiveness and conformity of mainstream Western science and academic work, Māori are less able to respond to deficit-oriented (re)presentations of Māori with strategies based on our unique ways of knowing and being. Including the extended facets of ourselves- our tūpuna in the natural environment who are present, living extensions of ourselves could be a way to include more strengths-based and aspirational narratives about Māori men in the research and the common discourse.

In the revised version of the article I have called the methodology conceptual in order to cover this, but can edit further if required.

l. 377 The  research is seen as  a way of storing knowledge --in Whanganui, through the protection  and ownership system called Kaiponu. The fact that  ‘new knowledge’ is not publically available raises questions of the research’s wider relevance:  are there ways in which the public and other research communities might share in and benefit by the research,  and under what the conditions  might this happen - eg by publication, open forum, community advice or wānanga?  Does this strategy for protection of intellectual property apply to your proposed research methodology (as well as your own research), and  are there occasions in which such protection will be relaxed?

The research certainly intends to share the new knowledge learnt through following this methodology so that other communities might benefit from it, however, the release of this knowledge will be conducted under the particular tikanga of Whanganui (Whanganuitanga) which includes Kaiponu. This is a step in the dissemination process which I have tightened in section three.

In the context of this article, I thought it would be useful to share upfront why Kaiponu came about in our region, and how this historical practice might be met in the research project. Kaiponu in the context of the research I cam conducting means that before I publish anything, I will seek the permission of my participants. To be clear, whilst I am publishing this article which is about my interpretation of Iwi knowledge in Te Awa Tupua, before any results or information containing the original knowledge is published, I will ask my participants what type of form would be most appropriate to publish as. So, yes, the research will be available to all those who may find it useful, but as our Whanganui tupuna did, the results will be coded in a dissemination form which is appropriate to hold that information safely. Anyone who wants to understand the information at a deeper level can approach Whanganui Iwi for further guidance and it will be at the Iwi’s discretion as to who this knowledge is shared with.

You have mapped some traditional categories used in empirical research (eg research questions, data collection, data analysis, self reflective journal) onto your practical application of korero tuku iho through summarising whakataukī . This suggests your model is situated between qualitative and quantitative research (but is mainly the former). Is it pertinent to introduce other interrelated aspects such as size of sample, organisation of data for analysis and presentation of results? Is there potential for revising the questions and direction of the research based on the data collection (i.e. the reflective journal)? What examples of ‘one size fits all’  (l. 565) conceptions of Indigenous research exist, against which your model is compared? 

I have ammeded the sentence to be clear about how important the variences in the research are to ensure the methodological approach can be adaptive to need. Yes, the reseach questions can certainly pivot, as long as the central objective of the research (to find evidence of strengths based narratives in Tāne Māori wellbeing practices) is met.

Reviewer 3 Report

Comments and Suggestions for Authors

Peer-review of : Kauaka e kōrero mō te awa, kōrero ki te awa

Kai ora, thank you for the opportunity to review this article. Overall, it is worth publishing because it makes a case for, and takes steps towards, a framework for research by Indigenous people (in this instance Māori from the Whanganui rohe) based on the belief that such research should be undertaken locally and guided by principles developed in contexts where the land and both the human and more-than human entities associated with it are in whakapapa relationships with one another.

The case is made that Whanganui iwi (and also other Indigenous peoples) have been exploited and misrepresented by researchers from outside their communities and that therefore iwi guardianship over information release should be maintained. There are some excellent descriptions of the worldview supporting this approach to developing research methodologies, such as the lines from 145-153 which explain how Indigenous cultures view the natural world as a living, indeed sentient, being whose perceptions humans can share through various means. As the author assumes, this article will probably have interest for other indigenous community members thinking about how to use research as part of desires to ‘live well’.

However, there are some revisions that would improve the structure of the article, enabling both those other researchers and the wider readership for the journal to have a clearer sense of what is being described and recommended. In particular, the announced focus on methodology is too diffuse in practice: the article seems to be moving forward on that subject but then detours or circles back frequently to sub-topics, creating the effect of foreshadowing, even promising, more than it delivers. That impression holds even given that the author says that they intend not to share some kinds of iwi knowledge, in accordance with the protective system and ethic of Kaipono, outlined in a section from line 325 onwards.

There are also a significant number of typos in the article as well as a problem with some prepositions and/or conjunctions pointing to the relationships between things, especially in the Introductory section.  Will give examples below. Since the key framing of whakapapa is based on relationships the small words that specify these relationships are important even though I imagine that this is easier to handle in te reo. A short section discussing how whakapapa relationships are connected linguistically in te reo Māori could be helpful – there are reference sources on this?

Numbers below are line references

30 connects humans deeply amongst? all other species (with? To? Not saying my options are correct but had trouble visualizing what was being referred to)

32 extends so deeply in? whakapapa (throughout?)

38 add (an) iwi’s genealogical connections …

40 Replace ‘To’ with For Ngati Rangi   (agrees better with  -‘ is referred to as a grandfather’)

49 and (as) psychic anchors

54 ‘revolve’ rather than ‘revolved’ since it is not just something in the past?

Good quote from Charles Royal

85- 86. These brief comments about Western ontologies and people supported by references to only two scholars are simplistic compared to the careful consideration given to Whanganui iwi. Have you followed the debate about the relationships between mātauranga Māori, Indigenous science, ‘western science’ etc. that started in the NZ Listener in the July 31, 2021 issue and is still ongoing? Whether or not you think that debate should ever have started, there are people from Māori, Pākehā and other ethnicities taking up all positions in that debate and many topics important to today's Aotearoa have been discussed there  and probably elsewhere as well] : can’t always predict all aspects of worldview on the basis of race.

102 Colonists and neo-colonists have used (even?) Western science as a weapon … don’t disagree with the assertion but not sure of purpose of the word ‘even’ .

108 Indigenous people around the world share a similar epistemology … had similar reaction as above to 85-86. Lot being assumed in one sentence. However you did pick up that topic again in more detail and more convincingly from 150 onwards in particular. This is an instance where could have been better to go straight into the evidence for your claim and not digress 110-115 into discussing Māori worldviews specifically.

145 – is the family name ‘Makiha’ rather than Mahika?

158-159 had to read sentence about the Yellowknives Dene a few times to figure it out. Would adding ‘is’ after Dogrib make it clearer? Taking Coulthard’s name out of brackets would help too leaving only (2010) in brackets since Coulthard is doing something (exemplifying) in the sentence not just being a reference. Putting too much in brackets happens a few other times too - e.g. 167 should be Wooltorton et al (2022) remind …. and 206 should be Macfarlane & Macfarlane (2019). Have never seen a sentence start with a bracketed reference.

173- 197 Good evocative examples and discussion. I wondered, before starting research planning or actual research is it recommended to go and physically talk/listen to the river (or other tupuna) or are they everywhere all the time and there is no need to do a physical action?

198. With the subheading ‘What might a whakapapa-based methodology look like? Felt like you were moving forward on a clear track but the next sub-heading 222 took a step back into describing the focus and planning of your Ph.D. How would this work if placed before the section that starts at 198?

223 Ngā Tāne Māori of Te Awa Tupua are introduced as both research participants and potential beneficiaries of research conducted in a tika way but the article never comes back to them, which leaves the points associated with them hanging as unresolved - had expected that research to be mentioned again in the Conclusion. For me, that research is associated with an unasked question – What might a whakapapa-based methodology be able to do? Bits of answers to that question are scattered throughout the article but not brought together. 

231 (a) deficit discourse (of) Māori men – ambiguous phrase. Who’s making that discourse? Is it them or others? Should the ‘of’ be ’about’ or ‘by’?

232. Should be ‘resilience’ rather than ‘resilient’?

244- Section on your own pre-ethics approach. Would that be stronger placed as your actualization of the Kaipono approach, which is also an ethical practice?

259 typo? The scared rope of Hinengākau

288 reconsider what it means to be human in multispecies environments … fascinating topic, do you return to it?

301 & following. Felt you were complexifying and qualifying in relation to the scope of the methodology when you still hadn’t really established what it was because you were wandering around the conceptual territory. However, noting that not all members of an iwi and neighbouring iwi have the same principles, values and knowledge is a good point (see my comments on 85 & 86).

318-324 good specific points

326- 343 is this section needed here? Maybe belongs with the Introductory paragraphs.

344 Discussion of Kaipono begins – could start here if that’s not too brutal. Strong section down to 360.

361-364. Quietly makes explicit that you are not going to present a clear methodology after all – ‘the how to is not as important as why’ … Should that intention be placed at the end of the Introduction so readers lower their expectations?

377 -380 feels like a short section taken from an earlier point in the process since you look forward to what you might or might not share (whereas by now the article should be fully delivering its main points rather than prefiguring what might happen later)

381 -383 ? this set of lines doesn’t connect with the sections before and after.?? Is it left over from an earlier draft?

384 –396 back to more general discussion

397 Earlier a list of all the kinds of knowledge that you could draw on had been given. In that list whakataukī and kōrero tuku iho were separate categories but this section says they are the same thing (without yet having given explained examples). Then 400-403 discusses using kōrero tuku iho in contemporary between-cultures settings. Doesn’t fit with the staunchness of your model that Indigenous and Western knowledges and practices are not compatible.

404 -568 This is the section where you finally  identify the possible bones of the methodology in terms of guiding principles for the conduct of research derived from whakataukī concerning Te Awa Tupua. The only suggestion here is to introduce the 4 terms/functions that you use to guide the unpacking of the whakataukī before you start the description and analysis.  However there are only 3 terms used: orientation, toolbox and lifecycle. What is number 4?   The section entitled Ngā manga iti, ngā manga nui e honohono kau ana, ka tupu hei awa tupua – doesn’t have one of these terms.

554-569 The Conclusion is very short and many thought-provoking topics introduced earlier are not referred to again, e.g. rsearch with Ngā Tāne Māori. If you agree that the structure could be re-shaped a little, then some of the cut material might be re-purposed in the conclusion. Perhaps the so far only implicit question What might a whakapapa-based methodology be able to do? Could be raised and discussed briefly in this section (i.e summarizing what you think the benefits would be) in the process inspiring other Indigenous researchers to respond.

Thanks for reading these comments. This is what I said to the editors:

“The article wanders around a fair bit; it sets out to explain an innovative methodology but there are many digressions on the way. The author is qualified to comment authoritatively on the material and the whakapapa/geneaological model is currently being applied in many areas of Māori scholarship. Some of the contextual discussions are necessary, vibrant and valuable but similar material occurs again later when it needn't. The proof of the argument comes late in the piece and then there is only a brief conclusion whereas some strong or intriguing points made earlier could be followed up on.

Have suggested a review of the structure and making more effort to group related points together so that the argument of the article can move forward more clearly, which need not involve giving away community-owned knowledge that the author wants to protect.

My recommendation is between minor and major revision but think it should be published after revision.

Comments on the Quality of English Language

English in the main is very good. There are a few typos and other errors. Some prepositions and conjunctions chosen occasionally lead to ambiguity. Have noted these occasions on the review file.

Author Response

Reviewer 3. Peer-review of : Kauaka e kōrero mō te awa, kōrero ki te awa (Genealogy Journal).

Kia Ora,

The article has been greatly improved thanks to your professionalism and insightful comments. I really value your input and appreciate the time and thought you put into offering suggestions for improvement. Your expertise added immense value to the article, and I am grateful for the detailed and constructive suggestions you offered. The article has undergone a restructuring process based on your feedback and that of the other three reviewers, incorporating the provided comments, to enhance its precision and focus.

I have made a list of the edits made below.

Once again, I appreciate your thoughtful participation. I hope that our academic paths will meet again in the future.

Ngā manaakitanga (kind greetings) from Aotearoa, New Zealand,

The author.

Edits made:

Comments on the Quality of English Language:

  • The references have been updated to “ACS” format

Please note that since editing this article, combining three other peer reviewers, line numbers may have changed slightly.

Comment

Addressed/response

In particular, the announced focus on methodology is too diffuse in practice: the article seems to be moving forward on that subject but then detours or circles back frequently to sub-topics, creating the effect of foreshadowing, even promising, more than it delivers. That impression holds even given that the author says that they intend not to share some kinds of iwi knowledge, in accordance with the protective system and ethic of Kaipono, outlined in a section from line 325 onwards.

This article seeks to illustrate the possibility for Indigenous communities to develop a research approach that aligns with their natural environments, place-based Indigenous worldviews (epistemologies), and community values, without necessitating detachment from their worldviews.

I have re-structured the article to demonstrate that this is a conceptual methodology presented here- sort of a niche dynamic work in progress, offered as guidance to fellow Indigenous researchers rather than over-promising a complete complex methodology.

I have broken the article into four sections for clarity and to set more (realistic?) expectations for the reader. 1- an introduction to an ancestral relationship with taiao, 2- relating to health and research 3- considerations a researcher might want to make in devising their own 4- sharing the conceptual methodology.

I have tried to be clearer about this article aim- to encourage researchers to explore their ancestral teachings, facilitating the creation of a methodological framework that integrates these teachings. This framework may encompass a range of research methods, whether derived from their own Indigenous practices or adapted Western methodologies, all grounded in the principles of their ancestral teachings. The article seemed to require much contextual information for a comprehensive and well-informed discussion, which I did not arrange very well- so I am very grateful for your suggestions. I think that like the river, I meander through several tributaries in writing this article so I am grateful for the clear navigational support.

There are also a significant number of typos in the article as well as a problem with some prepositions and/or conjunctions pointing to the relationships between things, especially in the Introductory section.  Will give examples below. Since the key framing of whakapapa is based on relationships the small words that specify these relationships are important even though I imagine that this is easier to handle in te reo. A short section discussing how whakapapa relationships are connected linguistically in te reo Māori could be helpful – there are reference sources on this?

Numbers below are line references:

30 connects humans deeply amongst? all other species (with? To? Not saying my options are correct but had trouble visualizing what was being referred to)

32 extends so deeply in? whakapapa (throughout?)

38 add (an) iwi’s genealogical connections …

40 Replace ‘To’ with For Ngati Rangi   (agrees better with  -‘ is referred to as a grandfather’)

49 and (as) psychic anchors

54 ‘revolve’ rather than ‘revolved’ since it is not just something in the past?

Good quote from Charles Royal

Thank you for identifying this. I think the prepositions are specific to Whanganui Iwi- and have been used intentionally to highlight the relational nature of Whanganui Iwi to the natural environment. I have restructured the article so that I introduce these linguistic ideas (in the form of kōrero tuku iho) in the hope that is more logical. I am happy to re-arrange the article further if required. If there are any examples where this does not make sense, I am happy to change them. Thank you for identifying this point.

I have amended these particular points below you have numbered- thank you.

85- 86. These brief comments about Western ontologies and people supported by references to only two scholars are simplistic compared to the careful consideration given to Whanganui iwi. Have you followed the debate about the relationships between mātauranga Māori, Indigenous science, ‘western science’ etc. that started in the NZ Listener in the July 31, 2021 issue and is still ongoing? Whether or not you think that debate should ever have started, there are people from Māori, Pākehā and other ethnicities taking up all positions in that debate and many topics important to today's Aotearoa have been discussed there  and probably elsewhere as well] : can’t always predict all aspects of worldview on the basis of race.

You have made an excellent suggestion here- and I had overlooked providing a much more nuanced commentary on this. I reallyu value this feedback. In this version of the article I have removed that discussion all together as the wordcount felt full, however I intend to write more substantively on this kaupapa in a second article.

Here is something I have written (not in the article) in reflecting on your point.

Recognising the importance of both Indigenous and Western scientific perspectives is critical, as is the fact that their cohabitation may contribute to a more comprehensive understanding of the world. This article privlidges Indigenous whakaaro (thinking) particular to Te Awa Tupua- as it is this very Indigenous knowledge which has historically been excluded from much research (particularly health research) discourse. Any amount of depth and careful consideration of the conceptual underpinnings of regional specific Mātauranga Māori (ancestral knowledge) is something I haven’t been able to find in depth in the academy, as it continues to exist inside of Indigenous communities. Yes, that conversation is important, but the focus of this article is on the application of mātauranga Māori rather than the Western Science vs Mātauranga Māori. This article attempts highlight how Mātauranga Māori to different Iwi is normal and valuable, so spends time exploring the applications of this notion in the hope it promotes a more varied and inclusive scientific conversation.

102 Colonists and neo-colonists have used (even?) Western science as a weapon … don’t disagree with the assertion but not sure of purpose of the word ‘even’ .

Thank you, another typo there, I have removed the word ‘even’

108 Indigenous people around the world share a similar epistemology … had similar reaction as above to 85-86. Lot being assumed in one sentence. However you did pick up that topic again in more detail and more convincingly from 150 onwards in particular. This is an instance where could have been better to go straight into the evidence for your claim and not digress 110-115 into discussing Māori worldviews specifically.

.

Thank you, I have moved the statement from 150 up to 85.

145 – is the family name ‘Makiha’ rather than Mahika?

Thank you- another typo, I have amended to Makiha.

158-159 had to read sentence about the Yellowknives Dene a few times to figure it out. Would adding ‘is’ after Dogrib make it clearer? Taking Coulthard’s name out of brackets would help too leaving only (2010) in brackets since Coulthard is doing something (exemplifying) in the sentence not just being a reference. Putting too much in brackets happens a few other times too - e.g. 167 should be Wooltorton et al (2022) remind …. and 206 should be Macfarlane & Macfarlane (2019). Have never seen a sentence start with a bracketed reference.

Thank you for identifying this, the brackets were an editing error with my End Note plug in. Brackets have been amended in the article. The referencing style has been amended to the ACS convention. Happy to edit further if required.

173- 197 Good evocative examples and discussion. I wondered, before starting research planning or actual research is it recommended to go and physically talk/listen to the river (or other tupuna) or are they everywhere all the time and there is no need to do a physical action?

I really like your question. It is up to the individual to work out what that means to them- as the title of the article suggests- a physical ‘going to’ the river and speaking with it might be a good idea. I had not used personal examples of how I engage with the river, but I personally go to the river physically for my engagement. I have added in the third section (considerations in devising a methodology) that I in fact did go to the Awa to converse before beginning the project. Thank you for reminding me to add something so obvious!

198. With the subheading ‘What might a whakapapa-based methodology look like? Felt like you were moving forward on a clear track but the next sub-heading 222 took a step back into describing the focus and planning of your Ph.D. How would this work if placed before the section that starts at 198?

I agree. I have re-arranged the article quite a lot, and I hope it does justice to your suggestion.

223 Ngā Tāne Māori of Te Awa Tupua are introduced as both research participants and potential beneficiaries of research conducted in a tika way but the article never comes back to them, which leaves the points associated with them hanging as unresolved - had expected that research to be mentioned again in the Conclusion. For me, that research is associated with an unasked question – What might a whakapapa-based methodology be able to do? Bits of answers to that question are scattered throughout the article but not brought together. 

A whakapapa based methodology might be able to provide a more appropriate way of conducting research with Tāne Māori, when their environment and ancestors are involved in the data collection and analysis process.

231 (a) deficit discourse (of) Māori men – ambiguous phrase. Who’s making that discourse? Is it them or others? Should the ‘of’ be ’about’ or ‘by’?

232. Should be ‘resilience’ rather than ‘resilient’?

I have ammended this, thank you

244- Section on your own pre-ethics approach. Would that be stronger placed as your actualization of the Kaipono approach, which is also an ethical practice?

I agree, I have moved it under the Kaipono section

259 typo? The scared rope of Hinengākau

Thank you for finding this, it has been amended

288 reconsider what it means to be human in multispecies environments … fascinating topic, do you return to it?

In the restructured article I have used it throughout to explain a Māori connection to taiao and have provided examples (eg. He Pou Herenga) to illustrate it. Happy to edit further if required.

301 & following. Felt you were complexifying and qualifying in relation to the scope of the methodology when you still hadn’t really established what it was because you were wandering around the conceptual territory. However, noting that not all members of an iwi and neighbouring iwi have the same principles, values and knowledge is a good point (see my comments on 85 & 86).

Thank you for pointing this out, I have expanded on that point in section three.

318-324 good specific points

326- 343 is this section needed here? Maybe belongs with the Introductory paragraphs.

I agree, I have restructured accordingly

344 Discussion of Kaipono begins – could start here if that’s not too brutal. Strong section down to 360.

I have shifted this section.

361-364. Quietly makes explicit that you are not going to present a clear methodology after all – ‘the how to is not as important as why’ … Should that intention be placed at the end of the Introduction so readers lower their expectations?

Thank you for pointing that out to me, I hope that I prefix that and set expectations a bit better in this latest version.

377 -380 feels like a short section taken from an earlier point in the process since you look forward to what you might or might not share (whereas by now the article should be fully delivering its main points rather than prefiguring what might happen later)

This is a good point. I had originally left this here as a sort of insurance clause to reassure a reader that Kaiponu would be used in the methodology (imagining local readers at home). However I see for this article is unnecessary in this section. I have removed this.

381 -383 ? this set of lines doesn’t connect with the sections before and after.?? Is it left over from an earlier draft?

Thank you for identifying this typo- you are correct, it was from a previous version. I have removed.

384 –396 back to more general discussion

I have shifted this in the restructure.

397 Earlier a list of all the kinds of knowledge that you could draw on had been given. In that list whakataukī and kōrero tuku iho were separate categories but this section says they are the same thing (without yet having given explained examples). Then 400-403 discusses using kōrero tuku iho in contemporary between-cultures settings. Doesn’t fit with the staunchness of your model that Indigenous and Western knowledges and practices are not compatible

Thank you for this comment. I have updated this paragraph for clarity:

·       This qualitative research drew from Whanganui-specific kōrero tuku iho (wisdom of our ancestors) including pūrākau (stories and narratives), tohu taiao (environmental observations), whakataukī/whakatauākī (expressions, idioms), interviews, grey literature, ruruku (incantation), tātai (short, codified chants directed to the environment) and kōrero a iwi (tribal specific teachings) to identify the sources of strength which nourish Tāne Māori.

·       Whakataukī/whakatauāki are a type of kōrero tuku iho, succinct messages which encapsulate and encode Tūpuna intelligence and transfer this to the living.

404 -568 This is the section where you finally  identify the possible bones of the methodology in terms of guiding principles for the conduct of research derived from whakataukī concerning Te Awa Tupua. The only suggestion here is to introduce the 4 terms/functions that you use to guide the unpacking of the whakataukī before you start the description and analysis.  However there are only 3 terms used: orientation, toolbox and lifecycle. What is number 4?   The section entitled Ngā manga iti, ngā manga nui e honohono kau ana, ka tupu hei awa tupua – doesn’t have one of these terms.

Thank you – this was a typo. I have added “the offerings” in its place.

554-569 The Conclusion is very short and many thought-provoking topics introduced earlier are not referred to again, e.g. rsearch with Ngā Tāne Māori. If you agree that the structure could be re-shaped a little, then some of the cut material might be re-purposed in the conclusion. Perhaps the so far only implicit question What might a whakapapa-based methodology be able to do? Could be raised and discussed briefly in this section (i.e summarizing what you think the benefits would be) in the process inspiring other Indigenous researchers to respond.

A whakapapa-based approach holds promise in fostering a deeper understanding of intergenerational connections, cultural heritage, and relationships within Indigenous families. This methodology has the potential to amplify the voices and perspectives of Indigenous communities, providing a more nuanced and authentic representation in research. Furthermore, it may contribute to the revitalization and preservation of traditional knowledge, offering a dynamic framework that resonates with the cultural values and epistemologies of Indigenous peoples. This discussion aims to not only explore the potential benefits of a whakapapa-based methodology but also inspire fellow Indigenous researchers to consider and engage with such approaches in their own work

Thanks for reading these comments. This is what I said to the editors:

“The article wanders around a fair bit; it sets out to explain an innovative methodology but there are many digressions on the way. The author is qualified to comment authoritatively on the material and the whakapapa/geneaological model is currently being applied in many areas of Māori scholarship. Some of the contextual discussions are necessary, vibrant and valuable but similar material occurs again later when it needn't. The proof of the argument comes late in the piece and then there is only a brief conclusion whereas some strong or intriguing points made earlier could be followed up on.

Have suggested a review of the structure and making more effort to group related points together so that the argument of the article can move forward more clearly, which need not involve giving away community-owned knowledge that the author wants to protect.

My recommendation is between minor and major revision but think it should be published after revision.

This has been so insightful. The first article I have ever written and I really appreciate your incredibly constructive feedback. Whilst I am still learning how to write academically (mostly the structure of writing) I found your feedback so helpful in looking at how to rewrite something as a bit of a compelling argument instead of my usual awa-styled flowly conversational style.

You have really helped me with your generous amount of feedback. Thank you! I am happy to rewrite this article again if you have further suggestions.

.

.

Reviewer 4 Report

Comments and Suggestions for Authors

This is an excellent article which requires very little work prior to publication. It offers, firstly, an elegant synthesis of current literature on the ways in which Māori engage with/relate to the environment and the natural world and how this shapes epistemological conceptions of history and of knowing and being in the world. It then presents a compelling case for 'place-based' research methodology which serves to complicate current 'Western' views of mātauranga as being suitable, appropriate and relevant for all iwi Māori. Instead, the author argues that attention to place, to the local and specific, drawing on the example of Te Whanganui awa (and taking an 'awa-centric' approach to understanding the world), needs to be progressed. 

The article is grounded in significant research and merits immediate publication (pending addressing the missing words as noted above). The Tupua Te Kawa framework is an example of how whakapapa and tuku iho might inform a more particular approach in conducting whakapapa-based research. A key benefit of this approach is that it avoids homogenising approaches to ‘Indigenous’ research methodologies.

The article is clearly written and has been well crafted; there does appear, however, to be a word or words missing on line 381. 

I hope the publication of this article and the methodology it proposes will inspire other Indigenous researchers to assume a similar approach and, piece by piece, further explore this methodology. 

Author Response

The article has been greatly improved thanks to your professionalism and insightful comments. I really value your input and appreciate the time and thought you put into offering suggestions for improvement. Your expertise added immense value to the article, and I am grateful for the detailed and constructive suggestions you offered. 

Edits:

- On line 381 I have completed the sentence which must have been lost in formatting the document. Thank you for spotting this. 

Once again, I appreciate your thoughtful participation. I hope that our academic paths will meet again in the future.

Ngā manaakitanga (kind greetings) from Aotearoa, New Zealand,

The author

Round 2

Reviewer 2 Report

Comments and Suggestions for Authors

This is much improved in sharpening the aims of the article to move from providing a clear methodology to  considering an approach to bespoke and customised research methodologies,  that  may  be iwi based  according to  particular ecosystems  - this focus is conveyed more consistently throughout the article

Some  reorganisation of the material to provide more self contained, and carefully titled subsections provides greater cohesiveness. This includes  a revised section 3 that  gives approaches towards  an iwi based methodology. More explicit signposting such as the statement of procedure ll. 84-98 and the indigenous perspectives on Māori health and well being research ll. 245-261, in the ‘Whakapapa to Taiao’ section, and the justification for section 3 (ll. 327-329) strengthens the foundation for the  place-based research approach identified and outlined in the  last section of the article

In relation to specific points, most of these are answered in some way as  you establish in your response; for example in the general comment  (potentially relatable to my query about what kinds of research projects would the  research methodology be applied to) that well being is a desirable outcome  of  indigenous research.  A sentence in the abstract is revised to drop science but to include ‘ science’ in section  3  in the concept of  a methodology (l.321), and although  you could not  find particular evidence for  using ancestral knowledge as research methodology in other cultures,  comments about the  benefits of  interpenetration of the  natural environment and human  well being  (ll.  245-258) in research reinforce the value of this approach. As you point out  in section  3 , this  iwi place-based model is likely to be widely acknowledged, because similar processes ‘are taking place in other research methodologies’ (l. 331). I note the references to  uses of indigenous research in other cultures has been moved to section  2 and placed in relation to rivers and other geological sites as  sentient entities to give greater  coherence; although the use of ‘conceptual’ for  the subtitle for section 4 doesn’t really cover the question  about deficit it at least allows for some space to consider extended facets of  selfhood such as tūpuna in future thinking. The clarification about the dissemination of the research through Kaiponu is welcome, and the  shorter length of this section is an improvement. The  grammatical emendations and  stylistic  modifications are all line with  academic prose.

One query, line 300 refers to ‘the literature’:  is this oral literature of well being narratives as identified earlier in the paragraph or written critical studies? This might  be clarified

Author Response

Peer Review response. Reviewer 3 Round 2.

4.3.24

Journal: Genealogy (ISSN 2313-5778)

Manuscript ID: genealogy-2684308

Title: “Kauaka e kōrero mō te awa, kōrero ki te awa: An awa-led research methodology” (Don’t talk about the awa, talk with the awa).

I sincerely appreciate your dedicated effort in providing insightful feedback during the second round of review for my article in the journal. Thank you for your time, expertise, and commitment to enhancing the quality of my article through your thoughtful and detailed peer review.

The author.

Comment

Adressed/response

This is much improved in sharpening the aims of the article to move from providing a clear methodology to considering an approach to bespoke and customised research methodologies, that  may  be iwi ased  according to  particular ecosystems  - this focus is conveyed more consistently throughout the article

Upon reading your comments I tightened some of the section on kōrero tuku iho by ammending a reference to Mahuika to strengthen this argument.

I had added an example on l.140 to illustrate how this iwi based variance can look.

Some reorganisation of the material to provide more self contained, and carefully titled subsections provides greater cohesiveness. This includes  a revised section 3 that  gives approaches towards  an iwi based methodology. More explicit signposting such as the statement of procedure ll. 84-98 and the indigenous perspectives on Māori health and well being research ll. 245-261, in the ‘Whakapapa to Taiao’ section, and the justification for section 3 (ll. 327-329) strengthens the foundation for the  place-based research approach identified and outlined in the  last section of the article

In reading your comment I reflected on some material I removed to reduced the wordcount, and realised a previously removed quote from (Royal, 1998) would be better added on l.324 and better ‘sign post’ the articles move from whakapapa taiao into the presented methodology on l.450.

I hope this is acceptable.  

In relation to specific points, most of these are answered in some way as  you establish in your response; for example in the general comment  (potentially relatable to my query about what kinds of research projects would the  research methodology be applied to) that well being is a desirable outcome  of  indigenous research.  A sentence in the abstract is revised to drop science but to include ‘ science’ in section  3  in the concept of  a methodology (l.321), and although  you could not  find particular evidence for  using ancestral knowledge as research methodology in other cultures,  comments about the  benefits of  interpenetration of the  natural environment and human  well being  (ll.  245-258) in research reinforce the value of this approach. As you point out  in section  3 , this  iwi place-based model is likely to be widely acknowledged, because similar processes ‘are taking place in other research methodologies’ (l. 331). I note the references to  uses of indigenous research in other cultures has been moved to section  2 and placed in relation to rivers and other geological sites as  sentient entities to give greater  coherence; although the use of ‘conceptual’ for  the subtitle for section 4 doesn’t really cover the question  about deficit it at least allows for some space to consider extended facets of  selfhood such as tūpuna in future thinking. The clarification about the dissemination of the research through Kaiponu is welcome, and the  shorter length of this section is an improvement. The  grammatical emendations and  stylistic  modifications are all line with  academic prose.

Thank you. I tried to answer your questions (which may have been in a round-about kind of way!)

In response to conceptual vs. deficit – I had cited an article (Johnson, 2021) on l.298 which speaks about the deficit narratives perpetuated about Tāne Māori. Perhaps I should of picked that word up again toward the end of the article?

I have attempted to remedy this by adding a sentence in the conclusion on line

This approach results in distinctively formed outcomes that offer valuable insights and connections that would be unattainable with a non-iwi-focused strategy and serves as a countermeasure against deficit-focused research and challenges negative stereotypes and narratives surrounding Māori men.

I hope this is adequate. 

One query, line 300 refers to ‘the literature’:  is this oral literature of well being narratives as identified earlier in the paragraph or written critical studies? This might  be clarified

Thank you for identifying this. When writing this sentence I was referring to the oral narratives and conversations with humans and the more-than-human (not the deficit focused narratives on Tāne Māori wellbeing). However for simplicity I have amended the sentence to read:

The next section considers the process of revitalizing the tāngata-taiao relationship as guided by kōrero tuku iho, into a methodology for conducting research.

I hope this simpler sentence makes more sense to the reader.

Reviewer 3 Report

Comments and Suggestions for Authors

Thank you for being open to my comments and suggestions. Your responses to my comments were painstaking and thorough: it is enjoyable to have a sense of academic collegiality even if indirectly.

You have done a lot of useful work on this since I last read it! The structure of the article is better in that the argument now flows forward with only a few circular moments, as is natural for a river. The components are in a more effective order to take the reader along with the article and while there are some sections that are appropriately discursive (where you are justifying the need for an iwi-based methodology based tightly on place) there are also now clearer sections (especially on kaipono and the 4 whakatauki) that are comparatively more crisp and can be seen as methodological by non-indigenous researchers also. There is just enough on your Tane Māori project to overcome the sense I had last time that something was missing - although in relation to the concluding reference to Tane Māori it would still be good to have a statement along the lines of for instance  'and this allowed us as researcher and research participants to do ....X & Y in a way [findings and/or dissemination/community effects, and/or sense of satisfaction] that we would not have been able to do under a non-iwi-focused method of conducting the research'  [you would obviously use different words - just something to highlight the benefits of the method].

158 - still says the  scared rope of Hinengākau.

One disconcerting small thing still for me is your frequent use of a single - (hyphen) jammed up against the previous word and then with a space after it. Usually I would think this implies that there will be another hyphen after you have finished with the idea/phrase that you are giving special attention to, but this doesn't happen.  I think you are probably using it instead of a semi-colon; to show you are now moving on to a slightly different idea. Even centering the hyphen would help not to trip up the reader's eye.

511 - are you writing about a Maramataka journal? If so, best not to split those words with a bracket in between.

597 How about replacing 'For this Ph.D project methodology ..' with something like 'Reflecting on this place-based methodology ....  By submitting to a journal you have moved on from the Ph.D stage and have the right to be more confident in your work.

So, generally I would be happy to see this published now, assuming it meets the publisher's guidelines for word-length and formatting (there was one paragraph near the end with several different fonts in it 524-532). If you need to cut back there are 2 short sections 363-366 and 496-497 that could be cut because similar points are made elsewhere.

Let it find some readers and see how they react. All the best]

Comments on the Quality of English Language

English generally good but a final editor's check would be beneficial.

Author Response

Comment

Adressed/response

Thank you for being open to my comments and suggestions. Your responses to my comments were painstaking and thorough: it is enjoyable to have a sense of academic collegiality even if indirectly.

Thank you so much for your direction- it really did help me write this article, and I enjoyed thinking about your prompts very much.

I have tried to (quite quickly) respond to your edits, but I am more than happy to tighten the article further if needed.

158 - still says the  scared rope of Hinengākau.

I have changed this to:

Whilst the Whanganui Awa is metaphorically named “Te Taura-whiri-a-Hinengākau” (the plaited rope of Hinengākau), which hints at the multi-layered understandings of ancestral worlds (Wilson 2022)

Is that what you meant?

One disconcerting small thing still for me is your frequent use of a single - (hyphen) jammed up against the previous word and then with a space after it. Usually I would think this implies that there will be another hyphen after you have finished with the idea/phrase that you are giving special attention to, but this doesn't happen.  I think you are probably using it instead of a semi-colon; to show you are now moving on to a slightly different idea. Even centering the hyphen would help not to trip up the reader's eye.

I have gone through the document with my grammar hat on (it is not a very well sued hat) and removed the single hyphen and replaced with commas and semicolons where I thought it made sense. Thank you for this suggestion.

597 How about replacing 'For this Ph.D project methodology ..' with something like 'Reflecting on this place-based methodology ....  By submitting to a journal you have moved on from the Ph.D stage and have the right to be more confident in your work.

Thank you. On l. 603 I have rephrased it to: “Together with the participants, I used this place-based methodology to share kōrero, which helped us comprehend the results and create solutions that are founded on our common worldview, and still provide room for other perspectives.”

511 - are you writing about a Maramataka journal? If so, best not to split those words with a bracket in between.

I agree, thank you- I have tidied that sentence up. This is in l. 516.

I developed my own Maramataka journal (a decision-making tool codified on the basis of Māori ecological knowledge based on the systematic study of environmental indicators, rhythms, and cycles) (Warbrick et al 2023)

You have done a lot of useful work on this since I last read it! The structure of the article is better in that the argument now flows forward with only a few circular moments, as is natural for a river. The components are in a more effective order to take the reader along with the article and while there are some sections that are appropriately discursive (where you are justifying the need for an iwi-based methodology based tightly on place) there are also now clearer sections (especially on kaipono and the 4 whakatauki) that are comparatively more crisp and can be seen as methodological by non-indigenous researchers also. There is just enough on your Tane Māori project to overcome the sense I had last time that something was missing - although in relation to the concluding reference to Tane Māori it would still be good to have a statement along the lines of for instance  'and this allowed us as researcher and research participants to do ....X & Y in a way [findings and/or dissemination/community effects, and/or sense of satisfaction] that we would not have been able to do under a non-iwi-focused method of conducting the research'  [you would obviously use different words - just something to highlight the benefits of the method].

I have attempted to do this, but it may still be quite vague and unclear. I have added in ll 320-324 a statement around why the methodology was devised in consideration of the participants (Tāne Māori of Te Awa Tupua):

If scholarly literature fails to adequately capture the worldview of Tāne Māori in my local context (Johnson 2021) exploring indigenous theorizing becomes imperative for research methodologies that are locally grounded and context specific. This encompasses recognizing and understanding their lived experiences with nature and entities beyond the human realm. To affirm Tāne Māori autonomy, providing a platform for Taiao (nature) and Tūpuna (ancestors) to articulate their teachings becomes pivotal within this framework.

I hope I have explained that here in the second to last paragraph in the conclusion section:

Together with the participants, I used this place-based methodology to share kōrero, which helped us comprehend the results and create solutions that are founded on our common worldview, and still provide room for other perspectives. I believe that the research participants serve as custodians of the ancestral wisdom inherent in Te Awa Tupua and practise their own kaiponu. The nuanced expressions of their varied and subtle identities requires the use of bespoke methodological approaches which are able to hold diverse voices and narratives that are aligned with their unique worldview which centres Taiao as a teacher, elder and source of wellbeing. These approaches are crucial for obtaining meaningful responses that align with the cultural context of Tāne Māori, and they seek to provide evidence of well-being practices for the future sons of Te Awa Tupua. By utilising the iwi-focused technique, I am able to incorporate their ongoing feedback during the dissemination process. This approach results in distinctively formed outcomes that offer valuable insights and connections that would be unattainable with a non-iwi-focused strategy. This positions Tāne Māori (the participants in the PhD research) as experts in their respective diverse contexts, embodying the profound traditions, values, and knowledge embedded in their experiences and expressions as descendants of a great river.

So, generally I would be happy to see this published now, assuming it meets the publisher's guidelines for word-length and formatting (there was one paragraph near the end with several different fonts in it 524-532). If you need to cut back there are 2 short sections 363-366 and 496-497 that could be cut because similar points are made elsewhere.

Thank you, the word count does seem high and I fixed the accidental font change in ll 524-536)

I did remove two sentences from the suggestions you identified thank you.

Let it find some readers and see how they react. All the best]

I am very grateful to you for your kind encouragement and expert lens on this article.

Ngā mihi maioha ki a koe. Thank you.